# Estimating Worldwide Impact of Low Physical Activity on Risk of Developing Ischemic Heart Disease-Related Disability: An Updated Search in the 2019 Global Health Data Exchange (GHDx)

**DOI:** 10.3390/medicines9110055

**Published:** 2022-11-03

**Authors:** Giuseppe Lippi, Fabian Sanchis-Gomar, Camilla Mattiuzzi, Carl J. Lavie

**Affiliations:** 1Section of Clinical Biochemistry and School of Medicine, University of Verona, Piazzale L.A. Scuro, 37134 Verona, Italy; 2Division of Cardiovascular Medicine, Stanford University School of Medicine, 450 Serra Mall, Stanford, CA 94305, USA; 3Service of Clinical Governance, Provincial Agency for Social and Sanitary Services (APSS), Via Alcide Degasperi, 38123 Trento, Italy; 4John Ochsner Heart and Vascular Institute, Ochsner Clinical School—The University of Queensland School of Medicine, 1401 Jefferson Hwy, Jefferson, LA 70121, USA

**Keywords:** physical activity, ischemic heart disease, epidemiology

## Abstract

We provide here updated analysis of the impact of physical inactivity on risk of developing ischemic heart disease (IHD)-related disability along with the latest 10-year progression. We collected data through an electronic search in the 2019 Global Health Data Exchange (GHDx) database using the keywords “low physical activity”, complemented with the additional epidemiologic variables “disability-adjusted life years” (DALYs; number); “ischemic heart disease”; “socio-demographic index” (SDI); “age”; “sex” and “year”, for calculating volume of DALYs lost due to physical activity (PA)-related disability after IHD (LPA-IHD impairment). Based on this search, the overall LPA-IHD impairment was estimated at 7.6 million DALYs in 2019 (3.9 and 3.7 million DALYs in males and females, respectively), thus representing nearly 50% of all PA-related disabilities. The highest impact of LPA-IHD impairment was observed in middle SDI countries, being the lowest in low SDI countries. The LPA-IHD DALYs increased by 17.5% in both sexes during the past 10 years (19.2% in males, and 15.8% in females, respectively), though this trend was dissimilar among different SDI areas, especially during the past two years. In high and high–middle SDI countries, the LPA-IHD grew during the past 2 years, whilst the trend remained stable or declined in other regions. In conclusion, LPA-IHD impairment remains substantial worldwide, leading the way to reinforce current policies aimed at increasing PA volume in the population.

## 1. Introduction

Cardiovascular disorders and, more specifically, ischemic heart disease (IHD) remain a highly prevalent healthcare issue worldwide due to the high burden of associated mortality and disability [1]. According to the heart disease and stroke statistics released in 2022 by the American Heart Association (AHA) [1], the prevalence of coronary heart disease is 7.2% in the US, slightly higher for males than females (i.e., 8.3% and 6.2%), with an estimated yearly incidence of acute myocardial infarction (AMI) of 3.7:1000, a substantial recurrence rate (i.e., 1-year recurrence: 5.3%), and accounting for a tremendous burden of direct and indirect healthcare expenditures (i.e., around $230 billion yearly). The chronic disability resulting from IHD is dramatically high since the rate of patients aged 45 years or older who will develop post-AMI heart failure is as high as 22% in females and 16% in males, respectively [1].

Physical activity (PA) has been conventionally defined as any body movement generated by skeletal muscles associated with significant increase in basal energy expenditure [1]. The first warning that an insufficient amount of PA may increase the risk of developing coronary heart disease was published by the AHA nearly 30 years ago [2]. They stated that regular physical exercise is needed to ameliorate the cardiovascular functional capacity, improve the lipid profile and reduce the impact of other well-known cardiovascular risk factors, such as overweight/obesity, diabetes and hypertension. This clear-cut advice has been reiterated by many other international organizations in the following years [3]. In 2020, for example, the World Health Organization (WHO) updated its guidelines on PA and sedentary behavior, emphasizing that adults should engage in a weekly program of 150–300 min of moderate intensity exercise, 75–150 min of high intensity PA, or equivalent combinations of these two [4], on the assumption of a curvilinear dose–response relationship between the volume of PA and the risk of death from cardiovascular disease. Identical indications are contained in the Physical Activity Guidelines for Americans [5], as well as in many other national PA recommendations and guidelines [6].

Epidemiologic evidence firmly supports the advice of augmenting physical fitness, whereby it was shown that middle-aged and older adults in particular can achieve a wide range of longevity benefits by increasing the level of PA [7]. Lee and coworkers demonstrated that physical inactivity (PI) causes 1 in 10 premature deaths worldwide, and that more than 1.3 million deaths worldwide could be avoided daily by decreasing PI by 25%. In addition, deaths related to PI are comparable to those due to smoking [8]. A comprehensive analysis carried out by Carlson and colleagues in 2018, using data from the National Health Interview Survey, concluded that as many as 8.3% of all deaths recorded in the US could be attributed to insufficient levels of PA among the general adult population [9]. Even more impressively, a meta-analysis based on 44 studies totaling 1,584,181 participants concluded that the risk of cardiovascular mortality was 23% and 27% lower in participants engaged in moderate and high leisure-time PA, respectively [10]. A linear negative association was also clearly observed between cardiovascular mortality and leisure-time PA, even after adjustment for multiple confounding factors. Another meta-analysis concluded that moderate to high PA could significantly lower the risk of immediate and 28-day mortality between 28–45% following the onset of an AMI [11]. Finally, a more recent overview of Cochrane systematic reviews, which evaluated the effectiveness of PA in reducing the risk of various clinical outcomes published by Posadzki et al. [12], showed a 13% reduction in the risk of death, compounded by significant quality of life improvement. Although this cumulative evidence leaves no residual doubts that PA should be considered an important part of a healthy lifestyle aimed at reducing the risk of developing IHD, its worldwide promotion remains suboptimal, thus representing an important cause of cardiovascular events/complications and human disability, as we previously showed with an analysis of the 2017 Global Health Data Exchange (GHDx) registry [13]. To this end, we aim to provide here an updated analysis of the impact of PI on the risk of developing IHD-related disability, based on the more recent 2019 GHDx database [14] and also according to its latest 10-year progression.

## 2. Materials and Methods

### 2.1. Study Design

The epidemiologic data on the impact of low PA levels on the risk of developing IHD-related disability were retrieved through an electronic search in the 2019 update of the GHDx database, a large worldwide repository of health-related information provided by the Institute for Health Metrics and Evaluation [15]. The GHDx is currently considered the most comprehensive worldwide repository of health-related data, made freely available for a vast array of health data research on an exhaustive list of diseases and injuries. By accessing this database according to the criteria defined in the following part of the manuscript, we aim to provide an updated analysis of the impact of PI on the risk of developing IHD-related disability, within a 10-year search period.

### 2.2. Data Source

We interrogated the GHDx database using the keyword “low physical activity” in the search field “risk”, complemented with the additional epidemiologic variables “measure” (set to “disability-adjusted life years”; DALYs); “metric” (set to “number”); “cause” (set to “ischemic heart disease”); location (set to “global”, “high SDI”, “medium–high SDI”, “middle SDI”, “low–middle SDI” and “low SDI”, where SDI stands for socio-demographic index); “age” (set to “+25 years”); “sex” (set to “both”, “males” and “females”); and “year” (year range between “2010 and 2019”), as summarized in Table 1 [14,15].

The SDI was used as a composite indicator of income per capita. Specifically, the SDI is a parameter that allows different countries to be classified based on their spectrum of development. It is typically expressed on a scale between 0 and 1; thus, it can be considered to be a composite measure of capita, education, and fertility rates. Based on these principles, the countries are classified as low SDI (SDI < 0.45); low–middle SDI (SDI, 0.45–0.61); middle SDI (SDI, 0.61–0.69); high–middle SDI (SDI, 0.69–0.81) and high SDI (SDI > 0.81) [15]. PA has been measured in total metabolic equivalents (METs) in the 2019 GHDx database and defined as average weekly PA (at home, work, transport-related, and/or recreational) < 3000–4500 MET minutes per week, whilst IHD has been defined as “disease of coronary arteries, usually from atherosclerosis, leading to myocardial infarction (MI) or ischemia, following the Fourth Universal Definition of MI and, for stable angina, physician diagnosis”, encompassing the associated International Classification of Diseases (ICD)-10 codes (Model estimation: DisMod-MR) [15].

### 2.3. Statistical Analysis

The data obtained from this multiple-domain search were then imported into a Microsoft Excel datasheet (Microsoft, Redmond, WA, USA), and were graphically plotted and statistically analyzed with Microsoft Excel. The study was performed in accordance with the Declaration of Helsinki and under the terms of relevant local legislation. This analysis was based on electronic searches in an open and publicly available repository (2019 GHDx), so no informed consent or Ethical Committee approvals were necessary.

## 3. Results

According to the GHDx database, IHD caused over 9 million deaths (first cause of death, overall) and 182 million DALYs (second cause of disability, overall) worldwide in 2019. The mean results of our analysis are shown in Figure 1. 

In 2019, the overall IHD-related impairment that could be attributed to low PA could be estimated at 7.6 million DALYs (3.9 and 3.7 million DALYs in males and females, respectively). Compared to the year 2010, the impact of IHD-related impairment that could be attributed to low PA (LPA-IHD impairment) slightly increased in both sexes (i.e., from 4.1 to 4.2% of all IHD-related DALYs), as well as in males (3.3 vs. 3.5% of all IHD-related DALYs), but remained instead virtually identical in females (5.2% vs. 5.2% of all IHD-related DALYs). In 2019, the LPA-IHD DALYs were around half of all DALYs caused by PA in both sexes (i.e., 7.6 vs. 15.7 million DALYs; 48.4%), as well as in males (i.e., 3.6 vs. 7.4 million DALYs; 52.7%) and females (i.e., 3.7 vs. 8.3 million DALYs; 44.6%), respectively). The highest impact of LPA-IHD impairment could be observed in middle SDI countries (2.5 million DALYs), followed by high–middle SDI countries (2.0 million DALYs), low–middle SDI countries (1.4 million DALYs), high SDI countries (1.2 million DALYs) and, finally, by low SDI countries (0.4 million DALYs). In high and high–middle SDI countries, the impact of LPA-IHD impairment was higher in females than in males, whilst the opposite could be seen in all the other countries.

The evolution of IHD-related impairment that could be attributed to low PA during the past decades is shown in Figure 2 and Figure 3. Globally, the LPA-IHD DALYs increased almost linearly until 2017, after which a plateau could be seen.

Overall, the LPA-IHD DALYs increased by 17.5% in the past 10 years (19.2% in males, and 15.8% in females, respectively) (Figure 2). Nonetheless, this trend was found to be considerably different among the different SDI areas. As shown in Figure 3, in both high and high–middle SDI countries, the LPA-IHD DALYs declined or remained almost stable until 2015, after which the trend started to increase again.

Unlike these areas, the IHD-related impairment that could be attributed to low PA constantly increased until 2017 in middle SDI countries, after which a plateau was apparently reached, whilst the LPA-IHD DALYs declined sharply after 2017 in middle–low and low SDI countries. As concerns the relative increase in LPA-IHD DALYs recorded during the past 10 years, this was found to be the highest in middle SDI countries (+26%), followed by low–middle (+23%) and low (+20) SDI countries, whilst the increasing trend was found to be substantially lower in high–middle (11%) and high (7%) SDI countries (Figure 2).

## 4. Discussion

The incidence of IHD is of such clinical relevance that the AHA has estimated that an American will develop an acute ischemic cardiac event nearly every 40 s [1]. Besides genetic predictors, a variety of demographical, metabolic, clinical and even environmental factors have been associated with increased risk of developing coronary artery disease, myocardial ischemia and a consequent functional impairment and/or chronic disability [1]. Currently, PI is recognized as a major health determinant across multiple biological and physical domains, since individuals engaged in regular physically activity display a reduced risk of developing a wide range of acute, and especially disabling, noncommunicable conditions such as cardiovascular disease, heart failure, osteoporosis, cancer, metabolic syndrome, diabetes, mental health disorders [16,17], as well as chronic migraine [18], and an enhanced likelihood of unfavorable COVID-19 progression [19], thus being responsible for a substantial social and economic burden worldwide [20]. Previous evidence emphasized that PI is dramatically high in certain countries, around 46% in the US [1] and 36% across Europe [21], so that the burden of IHD-related impairment attributed to low levels of PA remains substantially high [13], contributing to around 6–10% of the burden of a vast array of noncommunicable chronic diseases and premature deaths [22], and prompting us to conduct a specific analysis on the new 2019 GHDx database to provide more recent insights on this aspect.

According to our findings, the evidence that LPA-IHD DALYs were still considerably high in 2019, being half of all PA-attributable DALYs on human health, gives strong support to the evidence that the largest burden of disability caused by PA would directly encompass its interplay with the cardiovascular system, e.g., more specifically magnifying the individual risk of developing IDH. Some other interesting aspects have emerged from our analysis and deserve specific focus. First, the burden of LPA-IHD-related disability has certainly increased during the past few decades, displaying the highest (nearly exponential) growth from middle to low SDI countries, whilst the number of LPA-IHD has increased at a lower extent in middle to high SDI countries. This is perhaps the consequence of a combination of strengthened and more effective education policies towards the risk of PI in more developed nations during the past 7–8 years [23], but also by the fact that people living in middle to high SDI countries are engaged in lower volumes of daily PA (for their work), as recently highlighted by Guthold et al. in a pooled analysis of over 350 population-based surveys [24]. Therefore, major benefits are indeed expected by developing or promoting further PA recommendations in those countries where the volume of PI is higher in the general population.

However, a second important aspect that needs to be mentioned concerns the more recent evolution of LPA-IHD-related disability, which seems to differ broadly among different SDI areas. More specifically, although the global trend of LPA-IHD-related DALYs has apparently reached a plateau during the past 2 to 3 years (i.e., from our last analysis of the 2017 GHDx database), such a trend appears less evident in medium to high SDI countries, where the number of DALYs has instead continued to increase slowly. This would mean that the policies that have effectively convinced the people to engage in some forms of PA have somehow lost their efficacy in recent times in these countries. Conversely, a reversal trend has been clearly noted in low–middle and low SDI countries, whereby LPA-IHD-related disability has evidently declined during the past 2 years. Thus, although these nations will remain those with the sharpest increase in LPA-IHD-related disability during the past 7–8 years, their recent trend is promising, especially among women (Figure 3).

## 5. Conclusions

The impact of PI on the risk of developing IHD-related disability remains substantially high worldwide, thus leading the way to reinforcing current policies aimed at increasing awareness of this association as well as the volume of PA within the general population, especially in sedentary people. This emphasis is needed across all healthcare systems and governments worldwide, which should be actively engaged in providing guidance to caregivers for promoting PA. This goal can be achieved by means of reasonable and effective plans for active lifestyles to be broadly implemented for improving health and especially lowering the risk of developing cardiovascular disease, IHD and their detrimental, chronic disabling consequences [25].

## Figures and Tables

**Figure 1 medicines-09-00055-f001:**
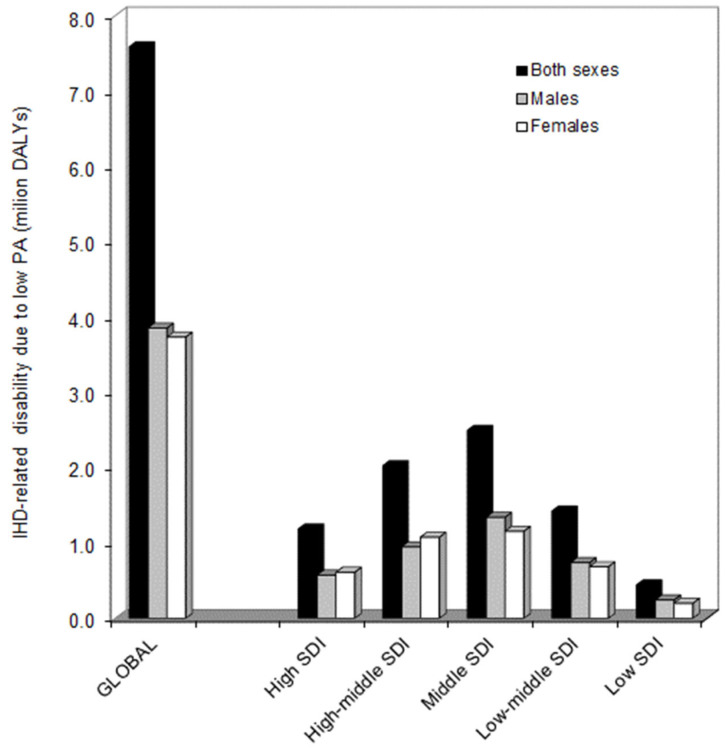
Ischemic heart disease-related impairment that could be attributed to low physical activity in 2019, based on data from the Global Health Data Exchange (GHDx) 2019 repository. DALYs, disability-adjusted life years; IHD, ischemic heart disease; PA, physical activity; SDI, socio-demographic index.

**Figure 2 medicines-09-00055-f002:**
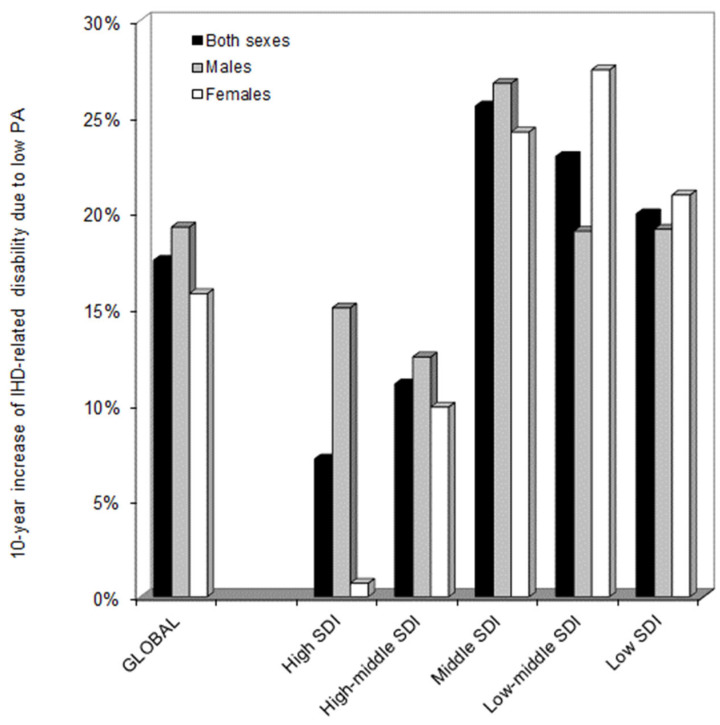
The 10-year relative increase in ischemic heart disease-related impairment that could be attributed to low physical activity in different socio-demographic index (SDI) areas based on data from the Global Health Data Exchange (GHDx) 2019 repository. IHD, ischemic heart disease; PA, physical activity; SDI, socio-demographic index.

**Figure 3 medicines-09-00055-f003:**
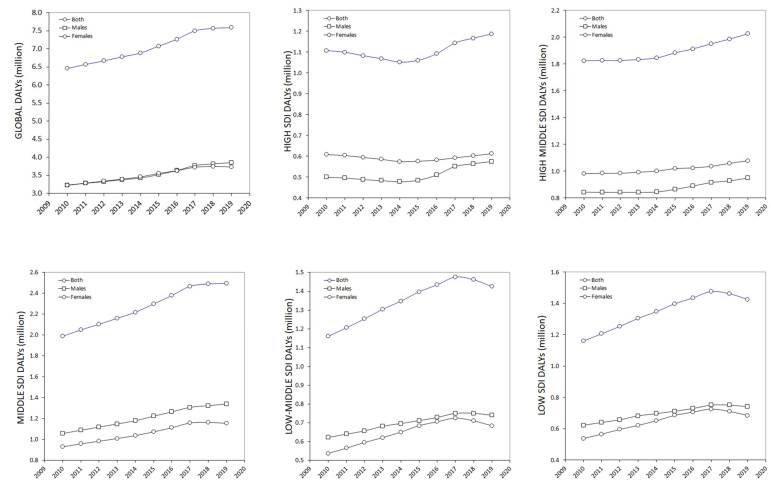
The 10-year evolution of ischemic heart disease-related impairment that could be attributed to low physical activity in different socio-demographic index (SDI) areas based on data from the Global Health Data Exchange (GHDx) 2019 repository. DALYs, disability-adjusted life years; IHD, ischemic heart disease; PA, physical activity; SDI, socio-demographic index.

**Table 1 medicines-09-00055-t001:** Search criteria used in the 2019 Global Health Data Exchange (GHDx) database [14,15].

Search Field	Search Term
Risk	“Low physical activity”
Measure	“DALYs”
Metric	“Number”
Cause	“Ischemic heart disease”
Location	“Global”; “High SDI”; “High–middle SDI”; “Middle SDI”; “Low–middle SDI”; “Low SDI”
Age	“+25 years”
Sex	“Both”; “Female”; Male”
Year	“2010”; “2011”; “2012”; “2013”; “2014”; “2015”; “2016”; “2017”; “2018”; “2019”

DALYs, disability-adjusted life years; SDI, socio-demographic index.

## Data Availability

Not applicable.

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
