# Peer review of "Estimating Worldwide Impact of Low Physical Activity on Risk of Developing Ischemic Heart Disease-Related Disability: An Updated Search in the 2019 Global Health Data Exchange (GHDx)"

_medicines, 2022, doi:10.3390/medicines9110055_

Round 1
Reviewer 1 Report (Previous Reviewer 1)
In the report titled “Estimating the worldwide impact of low physical activity on the risk of developing ischemic heart disease-related disability: 2019 GHDx update”, the authors have provided an updated analysis of the impact of physical inactivity or sedimentary behaviours on the risk of developing IHD-related disability. This study is based on the informational search on the 2019 GHDx database.
General comments:
The authors have provided and presented the information well. In my opinion, this paper is acceptable in its present form.
Author Response
We are thankful to the referee for the globally favourable comments on our manuscript. No action required.
Reviewer 2 Report (New Reviewer)
1. I have highlighted the reframed statements and suggested the same.
2. I just have one query, why is the methodology not described at all?
3. The study design can be mentioned in the title.

Author Response
- I have highlighted the reframed statements and suggested the same.
- ANSWER: We are really thankful to the referee for all these suggestions. Paper entirely revised as suggested (all new changes highlighted in yellow). As concern the question of the measure unit of DALYs, the measure unit is years lost, as in the definition.
- I just have one query, why is the methodology not described at all?
- ANSWER: Done, as suggested, as follows: “The epidemiologic data on the impact of low PA levels on the risk of developing IHD-related disability were retrieved through an electronic search in the 2019 update of GHDx database, a large worldwide repository of health-related information provided by the Institute for Health Metrics and Evaluation [15]” and “We interrogated the GHDx database using the keyword “low physical activity” in the search field “risk”, complemented with the additional epidemiologic variables “measure” (set to “disability-adjusted life years”; DALYs), “metric” (set to “number”), “cause” (set to “ischemic heart disease”), location (set to “global”, “high SDI”, “medium-high SDI”, “middle SDI”, “low-middle SDI” and “low SDI”, where SDI stands for socio-demographic index), “age” (set to “+25 years”), “sex” (set to “both”, “males” and “females”), and “year” (year range between “2010-2019”), as summarized in table 1 [14,15]”
- The study design can be mentioned in the title.
- ANSWER: Done, as suggested (i.e., “update search”).
This manuscript is a resubmission of an earlier submission. The following is a list of the peer review reports and author responses from that submission.
Round 1
Reviewer 1 Report
In the report titled “Estimating the worldwide impact of low physical activity on the risk of developing ischemic heart disease-related disability: 2019 GHDx update”, the authors have provided an updated analysis of the impact of physical inactivity or sedimentary behaviours on the risk of developing IHD-related disability. This study is based on the informational search on the 2019 GHDx database.
General comments:
The authors have provided and presented the information well. The report is well written and easy to follow for specialists. In my opinion, it is suitable for publication in this journal.
Minor comments:
1. The authors have written the manuscript nicely, but in this short report, the authors have used a lot of abbreviations which breaks the flow of the reading. If possible, the authors can reduce the number of abbreviations.
2. The authors should provide a brief introduction on the SDI with special mention of the different categories: “global”, “high SDI”, “medium-high SDI”, “middle SDI”, “low- middle SDI” and “low SDI”.
3. Figure 3 is not clear and hard to read. Authors should arrange the figure in a symmetric fashion, and the axis labels should be clearly visible.
Author Response
In the report titled “Estimating the worldwide impact of low physical activity on the risk of developing ischemic heart disease-related disability: 2019 GHDx update”, the authors have provided an updated analysis of the impact of physical inactivity or sedimentary behaviours on the risk of developing IHD-related disability. This study is based on the informational search on the 2019 GHDx database. The authors have provided and presented the information well. The report is well written and easy to follow for specialists. In my opinion, it is suitable for publication in this journal.
- We are thankful to the referee for the globally favourable comments on our manuscript. We’ll do our best to improve it according to the referee’s suggestions.
- The authors have written the manuscript nicely, but in this short report, the authors have used a lot of abbreviations which breaks the flow of the reading. If possible, the authors can reduce the number of abbreviations.
- ANSWER: Good point, thanks. Many abbreviations, all those unnecessary, have been eliminated and given in extension in the text.
- The authors should provide a brief introduction on the SDI with special mention of the different categories: “global”, “high SDI”, “medium-high SDI”, “middle SDI”, “low- middle SDI” and “low SDI”.
- ANSWER: This is a very good point, actually, the SDI is a summary measure that identifies where countries or other geographic areas sit on the spectrum of development. Expressed on a scale of 0 to 1, SDI is a composite average of the rankings of the incomes per capita, average educational attainment, and fertility rates of all areas in the GBD study. The SDI Quintiles are as follows: Low SDI: lower bound: 0 upper bound: 0.454743; Low-middle SDI: lower bound: 0.454743 upper bound: 0.607679; Middle SDI: lower bound: 0.607679 upper bound:0.689504; High-middle SDI: lower bound: 0.689504 upper bound: 0.805129; High SDI: lower bound: 0.805129 upper bound: 1. This clarification has been included in the follows: “Specifically, the SDI is a measure allow to classify different countries based on their spectrum of development. It is typically expressed on a scale between 0 and 1, can hence be considered a composite measure of capita, education, fertility rates. Based on this principles, the countries are classified as Low SDI (SDI, <0.45), Low-middle SDI (SDI, 0.45-0.61), Middle SDI (SDI, 0.61-0.69), High-middle SDI (SDI, 0.69-0.81) and High SDI (SDI >0.81) [15]”.
- Figure 3 is not clear and hard to read. Authors should arrange the figure in a symmetric fashion, and the axis labels should be clearly visible.
- ANSWER: Done, as suggested.
Reviewer 2 Report
In my opinion the paper was not clear, readable and informative and will not provide a valuable source document for anyone requiring a primer to know and understand this issue. Numerous shortcomings in the sections Materials/Methods, Results and Discussion make this paper inadequate for publishing in its current form. If the authors are interested, some comments below could be of use to them in the future. Some comments:
- Lines 17-28: The methodology and results are not presented clearly in the Abstract.
- Lines 32-34: Data refers to the whole world, cite an appropriate reference.
- Lines 43-45: Cite an appropriate reference for the definition of physical activity.
- Lines 58-60: The previous paragraph already mentioned that there are more recent physical activity recommendations and guidelines (references 3-6). Ref. No 7 was published in 2009, you have stated a type of physical activity - fitness (regarding which there are no data in the results of this paper) in the context with longevity (`in the Aerobics Center Longitudinal Study`, ..., `low cardiorespiratory fitness accounts for about 16% of all deaths in both women and men in this population`), and not with ischemic heart disease. There is no point for this sentence and reference in this manuscript.
- Lines 60-66: In this sentence you do not state study data for ischemic heart disease but for `all deaths`.
- Line 85: Insert a new subsection `Study design`, with adequate information.
- Line 85: Insert a new subsection `Data source`. Define all variables precisely, with citations of appropriate references. Define `disability-adjusted life years`, with citations of appropriate references. Define `socio-demographic index`, with citations of appropriate references.
- Lines 99-102: For the definition of Socio Demographic Index cite an appropriate reference.
- Lines 102-104: For the definition of ischemic heart disease that is stated in this sentence provide an appropriate reference. But, since the paper is based on the analysis of the Global Burden of Disease Study data, for the definition of the ischemic heart disease it must be listed what they have in that database included as the variable `ischemic heart disease`, according to the list of diseases and the list of case definitions are brought into line with the World Health Organisation nomenclature according to the International Classification of Diseases and Related Health Problems.
- Lines 105-107: The listed information does not belong to the section Materials and Methods.
- Line 108: Add a new subsection `Statistical analysis`. Define all indicators that are used to present results. List which statistical methods were applied in the paper. State how the statistical significance of observed differences was assessed. State the method that was used to assess the changes in frequency of contribution of low physical activity, i.e. `10-year relative increase of ischemic heart disease-related impairment that could be attributed to low physical activity in different socio-demographic index`, along with stating the test used to assess the statistical significance of differences.
- Lines 144-162: The text describes `trend`, without it previously being mentioned anywhere in the paper how the trend was determined, which test was used to determine whether it is statistically significant, etc.
- Lines 164-211: The discussion entirely is not in line with the aim of this paper, or with the good practice to compare the presented results with results of different similar studies in the world. Also, a potential explanation for the presented results is not provided, for differences between countries with different Socio Demographic Index, etc.
- Line 211: This manuscript has a lot of limitations. Add a new subsection `Limitations of this study`, list the limitations of the study with an explanation of why these limitations were not addressed in a good manner.
- Lines 236-306: The references should be cited in line with the Instructions to authors.
Author Response
In my opinion the paper was not clear, readable and informative and will not provide a valuable source document for anyone requiring a primer to know and understand this issue. Numerous shortcomings in the sections Materials/Methods, Results and Discussion make this paper inadequate for publishing in its current form. If the authors are interested, some comments below could be of use to them in the future. Some comments:
- We are thankful to the referee for the globally favourable comments on our manuscript. We’ll do our best to improve it according to the referee’s suggestions.
Lines 17-28: The methodology and results are not presented clearly in the Abstract.
- ANSWER: Good point, thanks, abstract revised accordingly: “We collected information through an electronic search in the 2019 Global Health Data Exchange (GHDx) database using the keywords “low physical activity”, complemented with additional epidemiologic variables (“disability-adjusted life years” (DALYs); “number”, “ischemic heart disease”, “socio-demographic index”, “age”, “sex” and “year”), for calculating the volume of DALYs lost due to physical activity (PA)-related disability after IHD (LPA-IHD impairment). Based on this search, the overall LPA-IHD impairment was estimated in 2019 at 7.6 million DALYs, 3.9 in males and 3.7 in females, representing ~50% of all PA-related disabilities. The highest impact of LPA-IHD impairment was observed in middle socio-demographic index (SDI) countries, being the lowest in low SDI countries”.
Lines 32-34: Data refers to the whole world, cite an appropriate reference.
- ANSWER: Done, this same information was available in the AHA document.
Lines 43-45: Cite an appropriate reference for the definition of physical activity.
- ANSWER: Done, this same information was available in the AHA document
Lines 58-60: The previous paragraph already mentioned that there are more recent physical activity recommendations and guidelines (references 3-6). Ref. No 7 was published in 2009, you have stated a type of physical activity - fitness (regarding which there are no data in the results of this paper) in the context with longevity (`in the Aerobics Center Longitudinal Study`, ..., `low cardiorespiratory fitness accounts for about 16% of all deaths in both women and men in this population`), and not with ischemic heart disease. There is no point for this sentence and reference in this manuscript.
- ANSWER: Good point, than, this paragraph and reference has been replaced with “Epidemiologic evidence firmly supports these advices, whereby it was shown that especially middle aged and older adults can achieve a kaleidoscope of longevity benefits by increasing the burden of PA [7]”. The new reference is of 2019.
Lines 60-66: In this sentence you do not state study data for ischemic heart disease but for `all deaths`.
- ANSWER: Yes, and we want to keep the sentence like that to demonstrate the global favourable impact of physical activity. No changes will be made regarding this comment.
Line 85: Insert a new subsection `Study design`, with adequate information.
- ANSWER: Good point. Done, as suggested (new information: “The GHDx is currently considered the most comprehensive worldwide repository of health-related data, made freely available for a vast array of health data researches on an exhaustive list of diseases and injuries. By accessing this database according to the some criteria defined in the following part of the manuscript, we aim to provide an updated analysis of the impact of PI on the risk of developing IHD-related disability, within a 10-year search period”).
Line 85: Insert a new subsection `Data source`. Define all variables precisely, with citations of appropriate references. Define `disability-adjusted life years`, with citations of appropriate references. Define `socio-demographic index`, with citations of appropriate references.
- ANSWER: Done. Citations are obviously those of the GHDx database, were all the information were extracted. Therefore, we added a single citation since this is the only data source.
Lines 99-102: For the definition of Socio Demographic Index cite an appropriate reference.
- ANSWER: As before, we have specified the meaning and classification of SDI, but, again, the reference is always that of the GXDx database. These measure have been constructed by the GHDx operators and there is no possibility to cite other sources. Text modified as follows: Specifically, the SDI is a measure allow to classify different countries based on their spectrum of development. It is typically expressed on a scale between 0 and 1, can hence be considered a composite measure of capita, education, fertility rates. Based on this principles, the countries are classified as Low SDI (SDI, <0.45), Low-middle SDI (SDI, 0.45-0.61), Middle SDI (SDI, 0.61-0.69), High-middle SDI (SDI, 0.69-0.81) and High SDI (SDI >0.81) [15].
Lines 102-104: For the definition of ischemic heart disease that is stated in this sentence provide an appropriate reference. But, since the paper is based on the analysis of the Global Burden of Disease Study data, for the definition of the ischemic heart disease it must be listed what they have in that database included as the variable `ischemic heart disease`, according to the list of diseases and the list of case definitions are brought into line with the World Health Organisation nomenclature according to the International Classification of Diseases and Related Health Problems.
- ANSWER: Yes, good point. Done, as follows: “IHD has been classified in the GHDx database based on the third universal definition of myocardial infarction, including the following (International Classification of Diseases (ICD)-10 codes: I20-I21.6, I21.9-I25.9, Z82.4-Z82.49″ (Model estimation: DisMod-MR) [15]”.
Lines 105-107: The listed information does not belong to the section Materials and Methods.
- ANSWER: Good point, yes. Moved to the results.
Line 108: Add a new subsection `Statistical analysis`. Define all indicators that are used to present results. List which statistical methods were applied in the paper. State how the statistical significance of observed differences was assessed. State the method that was used to assess the changes in frequency of contribution of low physical activity, i.e. `10-year relative increase of ischemic heart disease-related impairment that could be attributed to low physical activity in different socio-demographic index`, along with stating the test used to assess the statistical significance of differences.
- ANSWER: Done, as follows. All the other information are already present in the paper and it seems useless to repeat here. Moreover, we did not use any statistical method (we only showed trends), so that so methods can be listed here.
Lines 144-162: The text describes `trend`, without it previously being mentioned anywhere in the paper how the trend was determined, which test was used to determine whether it is statistically significant, etc.
- ANSWER: “Trend” means exactly “trend”. We do not really understand what the referee says here, nor we feel that providing a description of the term “trend” is necessary in a scientific journal, where everybody will understand it. This comment is actually hard to understand.
Lines 164-211: The discussion entirely is not in line with the aim of this paper, or with the good practice to compare the presented results with results of different similar studies in the world. Also, a potential explanation for the presented results is not provided, for differences between countries with different Socio Demographic Index, etc.
- ANSWER: We definitely disagree with the referee, and we are not willing to change the discussion, which seems perfectly aligned with the results and our conclusions. Moreover, the first referee does not raise any issue about the discussion, so that we feel comfortable in keeping our line.
Line 211: This manuscript has a lot of limitations. Add a new subsection `Limitations of this study`, list the limitations of the study with an explanation of why these limitations were not addressed in a good manner.
- ANSWER: Including a specific paragraph for “limitations” is not possible for this type of submission, as for the instructions to the authors. Moreover, we feel that this may be unnecessary, since these are already listed in several parts of the manuscript.
Lines 236-306: The references should be cited in line with the Instructions to authors.
- ANSWER: This was an invited paper and we have been advised that the editorial office will re-edit the reference according to their style. Therefore, it is unnecessary for us to do so.
Round 2
Reviewer 2 Report
Dear Editor,
My initial assessment for this paper was rejection, still of course I provided a comprehensive list of numerous shortcomings of the paper so that the authors can benefit from the review and improve their paper. I have now received the paper for re-review. The authors refused to address most of the comments without any proper explanation, and with a display of a lack of knowledge regarding the most important parts of the manuscript, leaving the paper still with many issues in the general writing, issues regarding methodological aspects, presentation and interpretation of the results. Apart from many errors in the received responses, even the newly introduced sentences (in the revised manuscript) are erroneous, for example "...SDI can be considered a composite measure of capita, education, fertility rates", etc. In addition, in many instances the authors have wrongly interpreted the comments in order to avoid addressing them - just one of the examples is the question regarding how trend was determined and what test was used to determine whether it is statistically significant, to which the authors replied that "everybody knows what a trend is". Not only do such responses (this was not the only one) lack basic professionalism, they still leave yet another issue unaddressed in this paper. Therefore, I recommend rejection of this paper, as the readership of your esteemed journal looks forward to seeing only methodologically sound papers.